# Soil-Borne Nematodes: Impact in Agriculture and Livestock and Sustainable Strategies of Prevention and Control with Special Reference to the Use of Nematode Natural Enemies

**DOI:** 10.3390/pathogens11060640

**Published:** 2022-06-01

**Authors:** Pedro Mendoza-de Gives

**Affiliations:** National Centre for Disciplinary Research in Animal Health and Innocuity (CENID-SAI), Laboratory of Helminthology, National Institute for Research in Forestry, Agriculture and Livestock, INIFAP-SADER, Morelos 62550, Mexico; mendoza.pedro@inifap.gob.mx; Tel.: +52-7319-2860 (ext. 124)

**Keywords:** nematode antagonistic organisms, biocontrol, plant and animal plague, sustainable control, eco-friendly control strategies

## Abstract

Soil-borne parasitic nematodes cause severe deterioration in the health of crops and supply animals, leading to enormous economic losses in the agriculture and livestock industry worldwide. The traditional strategy to control these parasites has been based on chemically synthesised compounds with parasiticidal activity, e.g., pesticides and anthelmintic drugs, which have shown a negative impact on the environment. These compounds affect the soil’s beneficial microbiota and can also remain as toxic residues in agricultural crops, e.g., fruits and legumes, and in the case of animal products for human consumption, toxic residues can remain in milk, meat, and sub-products derived from the livestock industry. Other alternatives of control with much less negative environmental impact have been studied, and new strategies of control based on the use of natural nematode enemies have been proposed from a sustainable perspective. In this review, a general view of the problem caused by parasitic nematodes affecting the agriculture and livestock industry, traditional methods of control, and new strategies of control based on eco-friendly alternatives are briefly described, with a special focus on a group of natural nematode antagonists that have been recently explored with promising results against plagues of importance for agricultural and livestock production systems.

## 1. Introduction

### 1.1. Nematodes in Nature

Nematodes, also called roundworms, are considered the most abundant metazoan organisms on Earth. It is estimated that soil nematodes can be found ranging from 1 to 100 × 10^6^ individuals/m^2^ of soil, mainly in the upper soil layers living in water films and water-filled pore spaces in the soil [1]. Nematodes can be found in decomposed organic matter in soil and plant roots and in other organic-rich substrates [2,3]. In addition to terrestrial environments, nematodes have adapted to most ecosystems, including aquifer environments, i.e., freshwater [4] and marine systems [5,6] and even the most extreme conditions where survival is difficult, i.e., in the polar regions of the world [7] and extremely high-temperature conditions [8]. Soil nematodes have a wide range of relationships with microorganisms of other species. Parasitism and predation are common ways of life, and some nematodes can be parasites or predators [9,10]. Similarly, soil nematodes play an important role in the food chain since they serve as food for other organisms of different taxonomic groups, i.e., mites [11] or nematodes [12], and at the same time, they feed on other organisms, including fungi, bacteria, and microarthropods [13]; additionally, nematodes participate as biogeochemical cycle regulators and enhancers of vegetation dynamics [14]. However, after hundreds of thousands of years, nematodes have developed an extraordinary capability to adapt to other biological systems, and thus, they have become parasites of animals, plants, and human beings [9]. In agricultural systems, soil nematodes can be divided into three groups: (1) entomopathogenic nematodes that feed on insects; (2) free-living nematodes that feed on different microorganisms, i.e., bacteria, fungi, and other nematodes; and (3) plant-parasitic nematodes that feed on plant tissues [15]. Next, we will deal with soil-borne parasites, particularly parasitic nematodes of importance for plants and ruminants.

### 1.2. Plant-Parasitic Nematodes

Plant-parasitic nematodes are worm-like pathogens usually less than 1 mm long, feeding on plant tissues [16]. They have a wide range of host plants, including foliage plants, agronomic and vegetable crops, fruit and nut trees, turfgrass, and forest trees [17]. Phytonematodes alter normal root functions, reducing rooting volume, foraging, and the efficient use of water and nutrients [18]. In this way, they are responsible for severe losses in economically important crops [3] that threaten global food security [19]. Phytonematodes are unsegmented worms with cylindrical thread-like bodies that taper at both extremes. Females could be from a cylindrical shape to an elongated or pear-shaped body [20]. Most phytonematodes have a needle-like structure called a “stylet” at the oral cavity that they use to feed on and kill plant tissues, particularly those cells of the root system, as in the case of a highly pathogenic group called the root-knot nematodes [21].

Nematodes belong to the Phylum Nematoda (roundworms) and Class Secernentea [22]. According to their feeding habits, plant-parasitic nematodes can be represented in a general manner by two groups of nematodes: (1) ectoparasites, that feed on the epidermis, cortical cells and root-absorbent hairs, but they do not penetrate the plant roots, and (2) endoparasites, that penetrate into the roots and feed on the root’s inner cells [23,24,25,26].

It is estimated that over 4100 species of plant-parasitic nematodes have been identified [27]. Some soil edaphic and ecological factors, e.g., altitude, temperature, moisture, soil pH, nutrients, and soil patches, influence the presence of different genera and species of phytonematodes [28]. Likewise, the presence of other microorganism species in their microhabitats also influences their population dynamics [29,30]. Furthermore, some plants have developed natural defence mechanisms through specific resistance genes that protect them from different pests, including nematodes [31]. Some of the most common genera of nematodes in agricultural soils, their hosts, their methods of attack, and their symptoms are summarised in Table 1.

#### 1.2.1. Life Cycle

The life cycle in most plant-parasitic nematodes is a similarly complex process involving different stages, i.e., eggs and distinct free-living pre-parasitic stages living in the soil and parasitic stages living in host roots. There is a simple and easy way to understand the life cycle of plant-parasitic nematodes, and it can be divided into two stages: pre-parasitic and parasitic. The pre-parasitic stage corresponds to free-living stages, basically comprised of the second juvenile stage emerging from the eggs when they search for the host cell; meanwhile, the parasitic stage starts when the nematode starts to feed on host roots [48]. Nematode parasitic stages possess a stylet situated at the nematode mouth at the rear end of the body that is used to penetrate the root cells and intake food from the plant tissues. The juveniles of the second stage (J2) of the root-knot nematodes penetrate the root near the root tip and initiate intracellular migration towards the apical meristematic region [49].

In the case of cyst-forming nematodes, they penetrate the plant roots and carry out intracellular migration to eventually settle at the vascular cylinder, where they develop syncytial-feeding sites within their host roots. Syncytia grow by incorporating protoplasts from dead cells. These organs serve as unique nutrient resources for development and reproduction through biotrophic interactions [50]. Second-stage juveniles (J2) develop three evolutionary stages to eventually become an adult. Adult males abandon the roots near the soil, where they mate with females. Once females are fertilised, they produce a large number of eggs that stay in the female body, forming a cyst where they are protected. Finally, when females die, eggs containing the J2 stage hatch and free-living (J2) nematodes will search for a new root to continue their life cycle [51].

#### 1.2.2. Economic Impact

Nematode plagues are some of the most serious problems affecting agricultural production all over the world and are even considered a global food threat [52]. Plant-parasitic nematodes (PPNs) pose a serious threat to the quantitative and qualitative production of many economic crops worldwide. It is estimated that plant parasitic nematodes cause 12.3% of crop losses, which means USD 157 billion annually [53]. Due to their widespread and devastating effect on economically important crops, root-knot nematodes (*Meloidogyne* spp.) are considered the most important nematodes throughout the world [54]. Additionally, *Meloidogyne* spp. can modify the plants’ defences, increasing their susceptibility to other pathogens, e.g., bacteria and fungi, which results in higher yield losses [55].

#### 1.2.3. Traditional Chemical Control Using Pesticides and Other Strategies

Plant-parasitic nematodes are complex individuals that, over millions of years, have had to overcome natural barriers, and they have had to develop adaptive strategies to survive. In this context, nematode control is not easy, and eradicating this problem is out of our reach. Therefore, reducing the parasitic population along with the damage caused to crops is the most realistic expectation. The control of plant parasitic nematodes must be faced from distinct perspectives. Some of the most common methods of prevention and control of plant parasitic nematodes, as well as their advantages and drawbacks, are shown in Table 2.

### 1.3. Gastrointestinal Parasitic Nematodes

#### 1.3.1. Definition

Gastrointestinal parasitic nematodes (GIN) affecting ruminants are a group of cylindrical-/filiform-bodied, 0.25 to 3 mm long, non-segmented worms, living as adult parasites in the gastrointestinal tract of animals or as free-living stages, e.g., eggs, pre-parasitic larvae, and infective larvae (L3) in faecal matter of infected animals [66]. They are responsible for a severe deterioration in the health and productivity of small ruminants and livestock all over the world [67,68].

#### 1.3.2. Common Ruminant Parasitic Nematode Genera/Species and Their Hosts

A high variation in the spectrum of genera/species of nematodes can be found when comparing agroecosystems [69]. A number of biotic factors, e.g., soil microorganisms and life cycle duration of different genera/species, and abiotic factors, including rainfall, temperature, humidity, and other factors associated with climatic conditions, determine the prevalence of GIN species found in either small ruminant flocks or livestock herds worldwide [70,71,72]. Nematode populations are not static and are always under a continuous dynamic. Some of the most common genera/species of nematodes in small and large ruminants include the following: in cattle, *Haemonchus* spp., *Mecistocirrus digitatus*, *Cooperia* spp., *Ostertagia* spp., (*Teladorsagia* spp.), *Trichostrongylus* spp., *Nematodirus* spp., *Bunostomum* spp., *Strongyloides* spp., and *Trichuris* spp. [73,74,75]; in sheep and goats, *H*. *contortus*, *Teladorsagia circumcincta*, *T*. *axei*, *T*. *comubriformis*, *T*. *vitrines*, *T*. *rugatus*, *Cooperia curticei*, *Nematodirus spathiger*, *N*. *filicollis*, *B*. *trigonocephalum*, *Oesophagostomum columbianum*, *O*. *venulosum*, and *Chabertia ovina* [76]. Reports on the presence of some genera/species of gastrointestinal nematodes in different countries with different climate conditions and in different host species are shown in Table 3. The use of traditional specialised taxonomic identification keys is an invaluable tool for morphological identification of adult nematodes at necropsy [77,78,79] and L3 from faecal cultures [79,80], for epidemiological studies, for research proposes, and for establishing control strategies. However, advanced molecular methods, e.g., qPCR followed by a high resolution melting analysis of ITS-1, open other convenient, rapid, and reliable alternative methods for taxonomic affiliation [81]. Similarly, new molecular tools, e.g., DNA metabarcoding using only faecal samples, have been claimed to provide a non-invasive method for assessing parasitic nematode populations [82]. Nevertheless, due to the high cost and the relatively small amplicon length, this cannot be considered a cost-effect method.

#### 1.3.3. Life Cycle

The life cycle of ruminant parasitic nematodes can be divided into two stages: one external, also called “exogenous,” that occurs outside the animals, where eggs and free-living and infective larvae are in faeces, soil, or on grass leaves, and another phase called “internal or endogenous stage,” where infective larvae are consumed by the animals together with the contaminated pasture, and once larvae enter into the alimentary tract, they migrate following the digestive flux to eventually establish themselves in corresponding organs, e.g., abomasum or small or large intestines [93]. Once L3 are in the abomasum or guts, they penetrate the gastric mucosa, where they have two options for development: they can remain in situ and initiate an arrested inactive phase of development, called hypobiosis [94], or they can continue with their subsequent stages of development, including histotrophic larvae or fourth larval stage (L4), pre-adult stages (also called L5) and eventually enter the sexually mature adult stage. When they reach the L4, they develop a small cavity at their rear end, equipped with a prominent needle-like structure called an “oral lancet,” which is specially designed to obtain blood from the stomach or gut veins, expressing their blood-sucking activity [95,96,97]. Adult males and females living in the digestive tract mate, and gravid females produce large amounts of eggs per day. It has been estimated that, in the case of *H*. *contortus*, around 10 thousand eggs per day are produced by a single gravid female [98]. The eggs of the parasites are expelled together with the faeces to the soil. When temperature and humidity conditions are favourable for the development of eggs, they develop L1 that hatch from the eggshell. The L1 grow and shed their external cuticles to transform into L2. This “ecdysis” process is repeated with L2 to eventually become in the subsequent larval stage (L3) [99]. A representative schematic diagram of the two phases of the life cycle of gastrointestinal parasitic nematodes is shown in Figure 1.

#### 1.3.4. Clinical Symptoms

Animals infected with parasitic nematodes can show a wide range of clinical symptoms, from very mild (almost imperceptible) to severe symptoms of disease and even the death of young animals [100]. Such severity in clinical symptoms depends on different epidemiological factors, such as body condition and origin of the animals, and host factors, including species, sex, and age [101]. Another crucial factor in expressing clinical symptoms in flocks is the number of infective larvae ingested in a short period [102]. In general, some of the most common symptoms reported for sheep and goats are weight and appetite loss, anaemia, weakness, and paleness of mucous membranes, mainly in ocular conjunctiva and subcutaneous oedema, i.e., jaw swelling, and diarrhoea [68]. Some symptoms can be attributed to specific genera/species; for instance, *Haemonchus contortus* can cause weakness, lethargy, lack of appetite, thirst, increased heart rate and breathing, pale conjunctiva and gingiva, and mushy stools in lambs [103], or even sudden death where animals can be found dead without preliminary symptoms [104]. Similarly, species associations can express different clinical frames; for example, abomasal nematodes, e.g., *Haemonchus contortus* and *Teladorsagia circumcincta*, can cause gastritis with severe clinical symptoms of anaemia and malnutrition in small ruminants. *Ostertagia ostertagi* and *Cooperia oncophora* were reported to be responsible for loss of appetite, scouring, and poor condition in cattle in the UK [105]. In general, small ruminants, particularly young kids, are more susceptible to being affected by nematodes than cattle [106]. However, a massive infestation with the abomasal nematode *Mecistocirrus digitatus* was reported as fatal for a cow in the Mexican tropics [73]. Productive parameters in small and large ruminants can be indicators of nematodiasis, for example, weight loss or milk production decreasing in dairy ewes [107].

#### 1.3.5. Economic Impact

The economic impact associated with GIN is enormous and continuous; since the life cycles of parasites are successfully completed every production cycle, and although deworming methods with chemical anthelmintics help to exert some control, this effect is only temporary because flocks and herds are always exposed to re-infection while they pasture. Thus far, no study has evaluated the global losses attributed to the effect of GIN. However, in different countries, the economic consequences of GIN in the livestock industry have been published. For example, in the United Kingdom, the cost of parasitic nematodosis in sheep was estimated at about GBP 99 million per year [108]. In Tunes, an average decrease in milk production and organ condemnation in cattle due to parasitic infections was estimated at 1.16 L animal^−1^ day^−1^ and 12.95%, respectively [109]. In another study, the annual costs of treatments against *H*. *contortus* were estimated at USD 46 and 103 million in South Africa and India, respectively [110]. Similarly, in Mexico, the economic impact caused by gastrointestinal parasitic nematodes in cattle was estimated at USD 445.10 per year [67]. However, in a recent study in 18 countries of the European Union, the average annual estimated cost of gastrointestinal nematode infections resistant to macrocyclic lactones was GBP 941 million for dairy cattle, GBP 423 million for beef cattle, GBP 151 million for dairy sheep, GBP 206 million for meat sheep, and GBP 86 million for dairy goats [111]. These are only some examples of the worrying and growing problems caused by GIN in the global livestock industry.

#### 1.3.6. Common Practices of Control and Their Advantages and Disadvantages

Traditionally, the regular administration of chemical anthelmintic drugs in flocks and herds is the most common practice of control worldwide. This method is very attractive for farmers because an effective and rapid lethal effect of these drugs against most parasitic nematodes is expected. However, the imminent development of anthelmintic resistance in parasites has dramatically diminished the efficacy of most commercially available anthelmintic drugs. This disadvantage is a limitation in the use of anthelmintic drugs since anthelmintic resistance is rapidly spreading, threatening the health of livestock [112,113,114]. However, using chemical anthelmintic drugs has other problems; for example, the presence of drug residues in animal products and subproducts, e.g., meat, milk, and derivates for human consumption, which is a potential risk to public health [115]. Anthelmintic chemotherapy causes other worrying problems, such as soil contamination and its implications for soil microorganisms, since anthelmintic drug residues are eliminated in the soil by the animals through faeces and urine. Some compounds are not degraded in the animals, and they remain in the soil as active molecules, affecting the microbiota and causing soil erosion in the long term [116,117,118]. The good news is that the use of chemical dewormers is not the only method of controlling ruminant parasitic nematodes. Several practices of control for gastrointestinal parasitic nematodes have been proposed, and they can be used as strategies to attack different evolutionary stages of the parasites according to their status in the parasitic life cycle [119]. In this review, we briefly summarise the most important strategies for controlling gastrointestinal nematodosis in small and large ruminants.

## 2. Biological Control

### Definition

The term “biological control” can be defined as the use of animals, fungi, or other microbes to feed upon, parasitise, or otherwise interfere with a targeted pest species [120]. In other words, biological control can be understood when human beings identify some antagonistic organisms in nature. The natural control agent is highly specific to an organism and is harmful to plants, animals, or human beings. This natural control system can be used to reduce the population of the undesirable organism using a natural enemy organism.

## 3. Natural Antagonists of Nematodes

### 3.1. Bacteria

In nature, bacteria and nematodes are closely related members of the soil biota. Different ecological interactions are established between these two types of organisms since they not only share the same microhabitat but also participate in the same ecological roles, such as food chains. Some nematodes mainly feed on bacteria [121]; in contrast, some bacteria are natural nematode killers, synthesising toxic, antibiotic, or inhibitory products of soil nematodes and acting as soil nematode regulators in nature [122]. There are many different bacterial species that use different and sophisticated physiological strategies to attack nematodes and eventually feed on them. One of the most widely studied nematode-antagonist bacteria is *Pasteuria penetrans*. This is a Gram (+) bacterium living in soil that produces endospores that attach to the nematode cuticle to penetrate it. They produce a large number of microcolonies inside the nematode body, and this invasion alters nematode reproduction [123] and eventually causes nematode death [124]. This bacterium has been found mainly as a parasite of the phytonematode *Meloidogyne incognita* and other phytopathogenic nematodes that affect the root systems of tomato and other economically important crops [125]. This bacterium has been found as a parasite of 323 nematode species belonging to 116 genera, including free-living nematodes, predatory phytonematodes, and entomopathogenic nematodes [126,127,128]. Several studies have revealed a potential use of *P*. *penetrans* for controlling root-knot nematodes, with an important implication on agricultural productivity. Unfortunately, this control system has not been shown to be a promising control method for ruminant parasitic nematodes since some attempts at *P*. *penetrans* spore attachment to the cuticle of ruminant parasitic nematodes were unsuccessful, and the inability of spores to attach to zoonematodes was demonstrated [129]. However, there are many records of the benefits of using *P*. *penetrans* in the control of root-knot nematodes. For example, in a study applying *P*. *penetrans* spores to *Meloidogyne arenaria*-infected tomato and oriental melon plants, the root gall numbers were significantly lower than in the control group [130]. In another study, the application of 3 x 10^5^ *P*. *penetrans* spores in a furrow revealed a reduction of up to 57.3% in *M*. *arenaria* egg production in the greenhouse [131]. These are just a couple of examples of the efficacy of this organism in the control of plant-parasitic nematodes. Many other genera/species of nematode-enemy bacteria have been explored as potential biocontrol agents of root-knot nematodes; these species include *Agrobacterium* sp., *Arthrobacter* sp., *Azotobacter* sp., *Clostridium* sp., *Desulfovibrio* sp., *Serratia* sp., *Burkholderia* sp., *Azospirillum* sp., *Bacillus* sp., *Chromobacterium* sp., and *Corynebacterium* sp. [132]. Other potential biocontrol agents with promising results are *Serratia plymuthica* [133], *Bacillus cereus* [134], *Pseudomonas fluorescens* [135,136], and *Bacillus thuringiensis* [137,138]. The use of natural nematode antagonistic bacteria is a promising sustainable alternative tool that could be used in the combined or integrated control of plagues caused by phytonematodes in economically important crops.

### 3.2. Protozoa

Although both protozoa and nematodes are very different organisms, they share the same soil habitat and are bacteria feeders. Protozoa are unicellular microorganisms living in soil, mainly in clay-rich soil, and most species feed on bacteria, similar to most free-living nematodes [139]. There is a lot of information about the ecological and physiological aspects of soil protozoa, including their prey role for predatory nematodes that feed on protozoa and other nematodes [140,141]; however, there is only limited information about the antagonistic and nematocidal effect of soil protozoa on nematodes. One common genus of soil flagellate, *Cercomonas* spp., attacks and kills the free-living nematode *Caenorhabditis elegans*, which is a much larger organism than flagellates. Once nematodes were added to a dense culture of flagellates, these protozoa attached to the nematode cuticle, and an increasing number of flagellates attacked the nematode cuticle. Some of them attached to the head and tail regions, exhausting the nematode and eventually killing it. Sometimes, flagellates enter the nematode’s body through natural orifices to invade and degrade their internal tissues [142]. In another study, a trophozoite of an isolate of the amoeba *Arachnula impatiens* was observed capturing a larva of *Meloidogyne incognita*, one of the most economically important nematode pests in agriculture. The amoeba captured the larva using fine filopodia and created several holes (2.5 to 5.5 μm in diameter) in the cuticular wall of the nematode and completely emptied its contents in only 3 h at 25 °C. Similarly, other amoeba identified as *Vampirella vorax* encysted after moving around the *M*. *incognita* larva body to eventually engulf and feed on the nematode within 12 to 24 h [143]. The complex adaptation process of protozoa and nematodes to their microenvironment has led them to develop mechanisms to attack and defend themselves from other microorganisms that compete against them for food, as is the case of the amoeba *Acanthamoeba castellanii* and the free-living nematode *C*. *elegans*. Both organisms are important microfauna predators in the soil, and both feed on bacteria. *Acanthamoeba castellanni* has developed an interesting metabolic strategy against that specific nematode species through the synthesis of exoproducts characterised as proteases and glycosidases with nematostatic activity and nematode repellent activity. Interestingly, *C*. *elegans* produces a biochemical counterpart that reduces the activity of these two enzymes as a defence response, and these exoproducts are also harmful to the amoeba, reducing their growth and increasing encystation. Thus, both microorganisms regulate their own populations in the soil [144]. In general, protozoa are important natural enemies of nematodes that deserve to be studied as potential tools of control for parasitic nematodes infecting plants and animals.

### 3.3. Acari

Acari are a subclass of ubiquitous arachnids with a tiny body, generally less than 1 mm in length with tracheal or cutaneous breeding, and many of them are parasites of other animals or plants. Acari, also called “mites,” inhabit both terrestrial and aquatic ecosystems and can even be found in relatively abundant amounts in aeroplankton [145]. Acari are one of the most diverse groups of arachnids on Earth, and about 60,000 species have been described [146]. Similar to the other organisms addressed in this review, mites have developed strategies to survive the adversities imposed by nature [147]. During this process, mites have established different bioecological associations with individuals of other taxonomic groups. In this way, mites are also involved in diverse biological activities, i.e., they are active members of food chains. For example, it is well known that mites are one of the favourite dishes for beetles. Beetles have a voracious appetite for mites. One adult beetle can eat 75 to 100 mites per day [148]. Similarly, mites feed on a wide variety of plant tissues, including economically important crops [149,150], and microorganisms, including bacteria, protozoa, algae, fungi, and nematodes [151,152,153]. The mite-feeding behaviour of nematodes was identified by Linford and Oliviera in 1934 and reported in 1938 [154], who reported that root-knot nematodes were devoured by soil mites. However, in another study, eighteen species of mesostigmatid mites were reported as nematode feeders. In this study, the combination of two different mites, *Caloglyphus* spp. and *Cephalobus* spp. (Nematoda), inoculated into the soil caused the nematodes to decrease in number [155]. In the case of animal-parasitic nematodes, some studies have shown that some mites, such as *Lasioseius peniciliger* and *Caloglyphus mycopagus*, have a voracious appetite for *Haemonchus contortus*, which is considered one of the most economically important ruminant parasitic nematodes, mainly in small ruminants [156]. Mites are potential candidates to be used as biological tools to control plant and animal parasitic nematodes; however, because of their wide range of food substrates, their living habits and their easy adaptive and invasive behaviour to many microenvironments make their management, use, control, and application difficult, and their behaviour should be deeply understood before they can be used in production systems.

### 3.4. Nematodes

Similar to those organisms previously addressed in this review, nematodes also have an important role in nature, mainly as important members of food chains, and they are also in charge of degrading organic matter and recycling nitrogen in the soil [28]. They are also important natural bioregulators of many other populations in the soil, including other nematodes. There is a simple way to categorise predatory nematodes based on their feeding apparatus and feeding behaviour in four major groups as follows: (a) Diplogasterids possess a large and strong buccal cavity with a strong claw-like movable dorsal tooth to grind their prey [157], Figure 2a; (b) Mononchids possess a highly sclerotised feeding apparatus with a large pointed dorsal tooth, small teeth, or denticles (Figure 2b); (c) Dorilaimids have a piercing and sucking system and use a needle-like feeding apparatus to puncture their prey and remove their contents; and (d) Aphelenchids are fungal-feeders, parasites of aerial plant parts, and insect parasites or predators, and the genus *Seinura* is the only member of the Aphelenchid group that are nematode predators. *Seinura* are small worms with a hollow stylet to puncture their prey and inject venom that is produced in an oesophageal gland [158]. Most studies about predatory nematodes have focused on the control of plant-parasitic nematodes, and only recently have some studies started to explore their antagonistic activity against ruminant parasitic nematodes. To be considered a good candidate as a predatory nematode, with high expectations as a potential agent of control of plant-parasitic nematodes, the predatory nematode should fulfil some characteristics, such as a good ability to search for its prey, be specific for its prey, be an efficient predatory nematode, have a certain life cycle duration and longevity, have reproductive potential, and be capable of surviving and adapting to ecological conditions [158]. The prey preference of some predatory nematode isolates is a good characteristic because it can focus control on a specific nematode prey. In a recent extensive revision, a large list of prey preferences for predatory nematodes is shown [158]. Although several studies with nematode–predatory nematodes have shown successful results under in vitro conditions, there are limited studies carried out under field conditions. The main limitations that researchers face in using predatory nematodes in the control of nematode plagues include mass culture and survival for a certain time after they are released in the field, which will have to be overcome for the management of either plant or animal parasitic nematodes.

### 3.5. Fungi

Fungi are one of the most abundant eukaryote groups of organisms on Earth, living in practically all types of ecosystems and substrates, including ocean sediments, salterns, rainforests, and even in the most extreme conditions, such as those in Antarctica [159]. Soil fungi share their microenvironment with a wide variety of organisms, including different kinds of nematodes, and consequently, different bio-ecological associations are established between fungi and nematodes, including competition, predation, and mutualism, with a tendency for equilibrium [30]. Fungi are one of the main natural nematode antagonistic groups and act as bio-regulators of nematode populations in the soil [160]. There is a large group of fungi that is considered the main natural enemy of nematodes, called nematophagous fungi [161]. Nematophagous fungi are cosmopolitan microfungi that attack or parasitise nematodes through different mechanisms. They can be classified according to their mode of action against nematodes in four groups: (a) toxin-producing fungi; (b) nematode-trapping fungi; (c) opportunistic or ovicidal fungi; and (d) endozoic or endoparasitic fungi [162]. These groups are briefly addressed next.

#### 3.5.1. Toxin-Producing Fungi

Some edible mushrooms have developed sophisticated mechanisms to kill nematodes; for example, they produce a powerful toxin that immobilises and shrinks the head of nematodes to infect them, kill them, and eventually digest their internal organs [163]. *Pleurotus* is one of the most widely studied the genus of edible mushrooms with nematocidal properties. *Pleurotus ostreatus* is a carnivorous fungus that preys on nematodes to obtain a nitrogen source in a nutrient-deficient environment. A nematocidal toxin obtained from *P*. *ostreatus* (NRRL 3526) immobilised 95% of the free-living nematode *Panagrellus redivivus*, and the nematodes never recovered. The toxin was identified as trans-2-decenedioic acid [164]. In a recent study, the mode of action of *P*. *ostreatus* demonstrated that the fungal toxin induces paralysis of prey nematodes via the cilia of nematode sensory neurons, followed by an excess of calcium influx and hypercontraction of the head and pharyngeal muscle cells to eventually cause necrosis and death of the nematode prey [165]. Other edible mushrooms, such as *Coprinus comatus*, form some structures called spiny balls that are responsible for paralysing and killing nematodes [166]. In another study, a hydroalcoholic extract obtained from the edible mushroom *Neolentinus ponderosus* showed potent in vitro and in vivo nematocidal activity using gerbils as a model of study against *H*. *contortus*, one of the most pathogenic parasitic nematodes affecting small ruminants worldwide [167]. This research showed evidence that secondary metabolites are present in edible mushrooms, and they should be further elucidated. Edible mushrooms can be considered potential alternatives for the control of plant and animal parasitic nematodes, although they still require further study to obtain a practical and functional biological control system.

#### 3.5.2. Nematode-Trapping Fungi

Nematode-trapping fungi are a group of microfungi living mainly in soil that are saprophytic organisms that retain nutrients from organic and decomposed matter in the soil; however, they have developed an extraordinary adaptation process to become predators of nematodes, with two facultative feeding alternatives [168]. The mechanism by which nematophagous fungi, after being saprobes, become predators of nematodes has been studied, and a morphogenesis inducer substance called “nemin,” a peptide produced by nematode cuticle peeling, is responsible for triggering the transformation of mycelia in trapping devices to be able to capture, penetrate, kill, and feed on the internal organs of nematodes [169] (Figure 3).

Nematode-trapping fungi produce different kinds of trapping devices, depending on their genus and species. Trapping devices can be classified as follows: (a) three-dimensional adhesive nets (Figure 4a); (b) constricting rings (Figure 4b); (c) simple or non-constricting rings (Figure 4c); (d) adhesive branches (Figure 4d); and (e) adhesive knobs (Figure 4e) [170].

The mechanisms of capture include trap formation, and this process is regularly accompanied by other sophisticated fungal strategies, for instance, the production of nematode attractant substances that mimic sexual and food olfactory cues [171]. Such predatory efficiency of nematophagous fungi has been demonstrated in a number of in vitro assays where both plant and animal-parasitic nematodes are captured in trapping devices and eventually killed and digested by predatory fungi [172,173]. Their potential use in the control of phytonematodes in different economically important crops has also been demonstrated with successful results [174,175,176]. The predatory activity of nematophagous fungi, both in vitro and in vivo, against animal parasitic nematodes has been widely documented [177,178,179]; some of the most studied nematophagous fungi in the control of plant and animal parasitic nematodes are species from the genera *Arthrobotrys*, *Duddingtonia*, *Purpureocillium* and *Pochonia chlamydosporia*. Some of the main characteristics of these fungi are briefly described.

##### Genus *Arthrobotrys*

The genus *Arthrobotrys* belongs to the order Orbiliales and family Orbiliaceae. To date, 71 species have been reported. Members of this genus are predatory fungi that capture, kill, and feed on nematodes. The genus *Arthrobotrys* was first reported by Corda in 1839 [180]. Then, its ability to act as a predatory fungus forming trapping devices to capture nematodes was reported by Zopf in 1888 [181]. Of the many species of this genus, *A*. *oligospora* is one of the most widely studied species. This species is one of the most abundant in nature, and it has been found throughout the world, living in most kinds of ecosystems [182,183]. As previously mentioned, depending on the *Arthrobotrys* species, they produce different trapping devices; for example, *A*. *javanica*, *A*. *vermicola*, *A*. *musiformis*, *A*. *superba*, *A*. *cladodes*, and *A*. *polycephala* produce adhesive nets, while *A*. *brochopaga* and *A*. *dactyloides* produce constricting rings [184,185]. The potential use of *A*. *oligospora* in the control of plant-parasitic nematodes has been assessed in economically important crops, for example, in tomato plants [173,186,187] and rice [188], which have been assessed in many trials, with promising results. In the case of using this species for controlling ruminant parasitic nematodes due to this fungus producing only a small amount of chlamydospores, mainly in old cultures, *A*. *oligospora* has not been considered a good candidate for use in the control of nematodes affecting ruminants. Although it produces a large number of spores, these spores are much more sensitive to the gastrointestinal passage of ruminants compared to *D*. *flagrans* chlamydospores, which are produced in large quantities in a spontaneous way [189]. Other important biological activities in *A*. *oligospora* have been discovered, including as a bio-regulator of metabolic processes in plants [190], lignolytic and cellulolytic activity [191], and potential biomedical activity as an immunostimulatory and antitumour agent [192].

##### Species *Duddingtonia flagrans*

The species *Duddingtonia flagrans* was first isolated in England by Dr C.L. Duddington in 1947 [193] from decaying vegetable matter. This fungus was reported as a new predatory species of *Trichotecium* that was seen as quite aggressive, capturing and killing nematodes. *D*. *flagrans* can produce three-dimensional adhesive nets, and it produces obovoidal to ellipsoidal conidia clusters regularly from groups of three to five from short, simple, and erect conidiophores with abundant chlamydospores [194,195]. Few records about the activity of this species against nematodes of importance for the agricultural industry have been published thus far [196,197]. However, in the case of the control of ruminant parasitic nematodes, several studies searching for the potential use of *D*. *flagrans* as a biological control agent have been published with excellent results [198,199,200]. The control of ruminant parasitic nematodosis using *D*. *flagrans* works due to the spontaneous production of large amounts of chlamydospores. These are thick-walled resistance spores produced to ensure their survival in adverse conditions [201]. Chlamydospores are produced in the lab to obtain sufficient fungal inoculum to be orally supplied to animals. Chlamydospores can be mixed with animal feed to be offered to the animals. Once animals ingest chlamydospores, these pass through the digestive tube and are eventually eliminated with the faeces to the soil. In this way, close contact between the nematode eggs and recently hatched larvae and the fungal chlamydospore takes place in situ. When nematodes develop from the first developing stage to the second and third evolutive stages, larvae eliminate some cuticular peeling cells. The peeling cell products include nemin, which possesses some binding receptors that establish contact with receptors present on the surface of the fungal cells. This binding stimulates the fungal cells to trigger the transformation of mycelia into trapping devices, which is a physiological process called “morphogenesis” [202]. Larvae are trapped, killed, and finally degraded and digested using the enzymatic strategies of the fungus [203]. This process provokes a blockade of the life cycle of the free-living stages of the parasite in the faeces and a substantial reduction in the larvae population in faeces [204]. Thus, a much lower quantity of infective larvae is spread to the grassland, and consequently, the animals consume a much lower number of infective larvae. This is an indirect biological control system since the infected animals consuming *D*. *flagrans* chlamydospores will maintain their parasitic burden; however, they will not be reinfected in the same way since pastures, which are the main source of contamination, will have less infective larvae after maintaining this biocontrol system [204]. This system of control has many advantages in comparison with the traditional methods of control using anthelmintic chemical drugs; for instance, it does not contaminate the environment, does not provoke resistance, and does not leave toxic drug residues in milk, meat, or sub-products of animal origin.

##### Species *Purpureocillium lilacinum* syn. *Paecilomyces lilacinus*

*Purpureocillium lilacinum* is a soil microfungus, first described by Bainier in 1907. It lives in different ecosystems, mostly tropical and subtropical soils [205], and is a common member of the soil saprophytic mycobiota. This species is a very peculiar cosmopolitan soil fungus, with extraordinary biological versatility and an enormous capability to adapt to many environmental conditions. *P*. *lilacinum* has been considered one of the most important natural enemies of pests of agricultural importance [206]. This species has been identified as a parasite of different kinds of plant pests, including nematodes, and it has been proposed as a biotechnological tool for controlling diseases caused by phytopathogenic nematodes affecting important commercial crops [207]. *P*. *lilacinum* is a well-known egg-parasitic nematode. This species can grow over the surface of the eggshell of root-knot nematodes and penetrate it, occasionally through the formation of an appressoria [208]. In addition to ovicidal activity, *P*. *lilacinum* has also been found to affect all life stages of the root-knot nematode *Meloidogyne incognita* using a similar strategy [209]. *Meloidogyne incognita* is one of the most common pathogenic nematodes, and it is responsible for devastating losses in agriculture worldwide [210]. In addition to its important nematocidal activity, *P*. *lilacinum* has also been found to be a natural enemy of plant pathogenic insects. For instance, this species has been reported as a biocontrol agent of many species of leaf-cutter ants, such as *Acromyrmex lundii* and Atta ants. This association has led to a reduction in the population of ants, and it helps to reduce the indiscriminate use of chemical pesticides [211]. However, in a recent study, the use of different entomopathogemic fungi, including *P*. *lilacinum*, showed high insecticide activity against termites, and an important beneficial impact in the management of soil micro-arthropods was recorded [212]. Regarding the activity of *P*. *lilacinum* against animal parasitic nematodes, little information has been documented thus far. The ovicidal activity of two strains of *P*. *lilacinum* against dog-parasitic nematode eggs (*Toxocara canis*) was recorded by Gortari et al. (2008) [213]. The activity of *P*. *lilacinum* against parasites of importance in the livestock industry has only been published in a few papers. In a recent study, the lethal activity of organic extracts obtained from different nematophagous fungi, including *P*. *lilacinum*, showed an important nematocidal effect against the larval infective stage of *Haemonchus contortus*, the most pathogenic zoonematode affecting small ruminants [214]. The use of *P*. *lilacinum* is a promising potential biotechnological tool of importance in the control of ruminant parasitic nematodes that deserves to be widely explored to replace, at least partially, the indiscriminate use of chemical anthelmintic drugs that risk public health and the environment and provoke resistance in parasites [204]. Additionally, *P*. *lilacinum* establishes biological associations with other organisms that involve it in important environmental roles. Such extraordinary versatility has led this species to enter plant tissues, becoming an endophytic organism that promotes plant defences against pathogens, e.g., reducing cotton aphid populations [215], and in other studies, it has been identified as an important bio-stimulant of plant growth and crop yield [216].

##### Species *Pochonia chlamydosporia*

*Pochonia chlamydosporia* belongs to the group of Hypocreales, a nematophagous fungus that used to be classified as a member of the genus *Verticillium*; however, when new nomenclature was proposed in 2012, it was reclassified as *P*. *chlamydosporia* [217]. This fungus is considered a multitrophic species since it has been identified as parasitic or pathogenic for invertebrate and nematode hosts [218]. It has also been considered an endophytic organism since it can penetrate the plant’s root tissues and play an important role in inducing plant resistance against phytopathogenic nematodes [219]. Similar to other nematophagous fungi, *P*. *chlamydosporia* is a saprophytic fungus and a facultative nematode parasitic fungus [220]. This species is well-known due to its extraordinary capability to colonise the surface of the nematode eggshell and to develop an appressoria to penetrate the egg and eventually feed on nematode embryonic cells [221]. This species is also capable of parasitising root-knot nematode females [222]. *P*. *chlamydosporia* has been assessed as a potential biotechnological tool for the control of plant plagues caused by root-knot nematodes [223,224] and against animal-parasitic nematodes [225,226]. In the case of using *P*. *chlamydosporia* in the control of animal-parasitic nematodes, the timing of egg hatching in faeces, followed by the emerging larvae, is so rapid (ranging from 3 to 5 days) [227] that it is not sufficient for *P*. *chlamydosporia* to colonise the faecal matter and exert its egg-parasitic activity before eggs become larvae. This should still be deeply studied to identify a successful strategy to use *P*. *chlamydosporia* to reduce GIN parasitic nematode eggs in faeces. However, studies on secondary metabolites in *P*. *chlamydosporia* have provided important information that offers a number of promising biotechnological compounds with potential use in the control of parasitic nematodes of importance in the livestock and agriculture industries [228]. In contrast, the use of *P*. *chlamydosporia* in the control of plant-parasitic nematodes should be widely explored as an additional tool in integrated systems of control of plagues of importance in agriculture according to the different crop production systems in different agroecological areas of the world.

## 4. Conclusions

The excessive use of chemical pesticides for controlling agricultural pests, particularly phytoparasitic nematodes, as well as the use of chemical anthelmintic drugs for controlling ruminant parasitic nematodes, are increasingly discredited strategies due to a number of undesirable consequences. After intensively using these chemical compounds, their final destiny remains as pollutants, either in soil or in aquifers, that severely affect the environment, particularly the soil microflora and microfauna, putting the stability of ecosystems at risk. Additionally, the anthelmintic drugs administered to animals can remain in animal tissues for human consumption, e.g., milk, meat, or sub-products. Similarly, chemical pesticides contaminate the environment with the same devastating effects on beneficial organisms in nature. Other drawbacks in the use of these chemical compounds are that parasitic nematodes, either phytonematodes or zoonematodes, after constantly being exposed to these molecules, develop mutations that allow them to overcome the lethal effect of the synthetic compounds; the selection of resistant pathogens exacerbates the problem. Nevertheless, there are other alternatives of control that can be used in integrated control programmes for agricultural pests or ruminant parasitic nematodes. As we have shown, plagues of importance in the agriculture or livestock industry are highly complex organisms, and they should be controlled using various strategic tools. In the present review, different strategies of control have been shown and briefly discussed, making special reference to the use of natural nematode antagonists that have been explored as potential tools of control from the perspective of sustainability. Some of the organisms mentioned in this review are still under basic study, and more information should be generated to consider them as practical measures of control. Other nematode enemies have already provided excellent biotechnological tools for the control of nematodes, affecting important economic crops and/or against ruminant parasitic nematodes. It is important to consider that research in this important area of knowledge should be encouraged since the intensive use of chemically synthesised molecules and their negative effects threaten the environment and public health all over the world.

## Figures and Tables

**Figure 1 pathogens-11-00640-f001:**
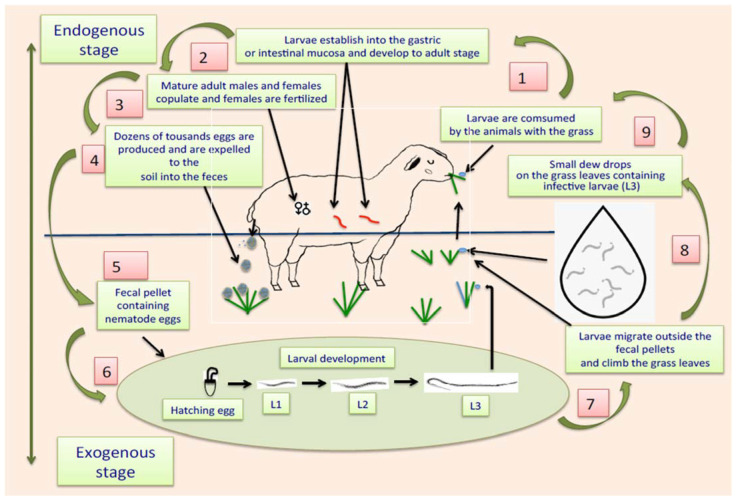
Scheme showing the life cycle of gastrointestinal parasitic nematodes affecting *ruminants*.

**Figure 2 pathogens-11-00640-f002:**
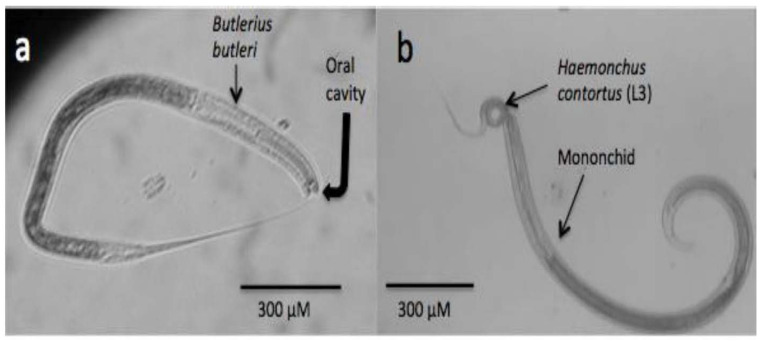
Microphotographs showing two different kinds of predatory nematodes. (**a**) *Bulterius bulteri*; (**b**) mononchid feeding on an infective larva (L3) of *Haemochus contortus*, a parasite of sheep and goats.

**Figure 3 pathogens-11-00640-f003:**
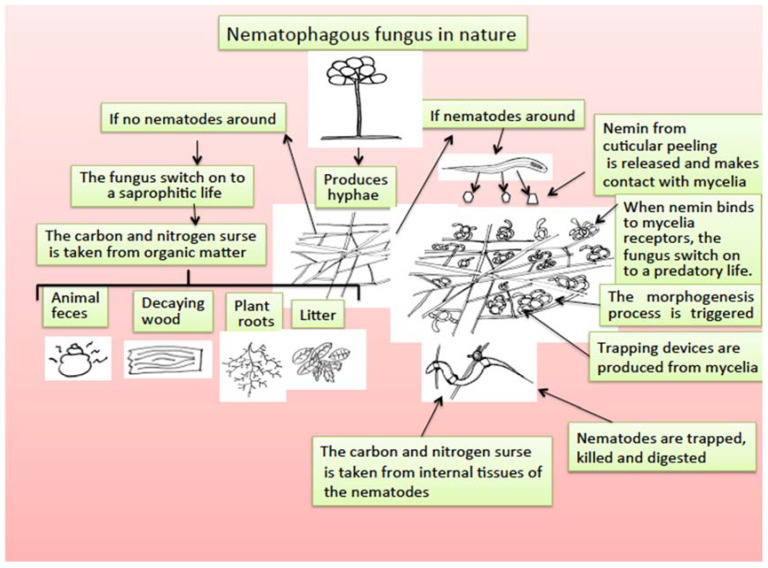
Scheme for nematophagous fungi duality to switch from saprophytic to a predatory life promoted by the presence of nemin from nematode cuticle peeling.

**Figure 4 pathogens-11-00640-f004:**
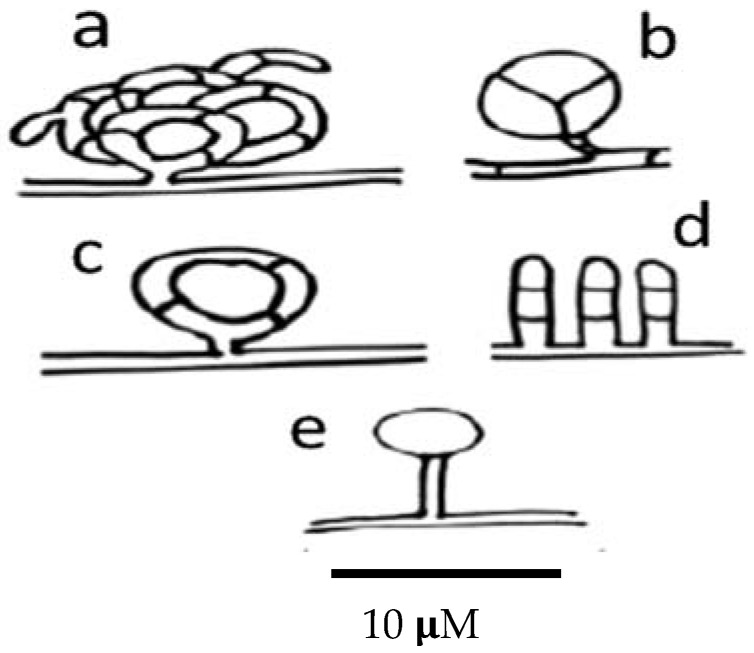
Scheme showing the different types of trapping devices produced by nematophagous fungi. (**a**) Three-dimensional adhesive net; (**b**) constricting ring; (**c**) simple ring; (**d**) adhesive columns; and (**e**) adhesive knobs.

**Table 1 pathogens-11-00640-t001:** Some of the most common genera of nematodes in agricultural soils, hosts, methods of attack, and symptoms.

Genus/Host Range	Plant/Crop Host	Method of Attack	Symptoms	Author
*Meloidogyne* spp. Root-knot nematodes More than 90 host species	Wide horticultural and field crop host range (about 2000 plant hosts worldwide)	Root system	Root galls Dead in young plants	[32]
*Nacobbus aberrans* False root-knot nematode	Affects a number of economically important crops, e.g., tomato, chilli pepper, beans, potatoes, sugar beets, and crucifers	Migratory/sedentary Endoparasitic nematode Penetrate into plant roots, forming galls	Root galls	[33,34]
*Aphelenchoides* spp. More than 200 species	Wide host spectrum, including ornamentals. Some species are fungi feeders	Some species endoparasitic in leaves, but also feeds ectoparasitically on leaves and flower buds in some plants	Chlorosis and necrosis of leaves	[35,36]
*Heterodera* spp. At least 80 species Obligate parasites Affects more than 40 species	A few hosts, including: oatmeal, soybean, alfalfa, corn, and others	Penetrate cortex roots, endodermis, or vascular parenchyma Feeds on root tissues	General debilitation Reduction in the efficiency of the root system Chlorosis, stunted growth, wilting Poor yield	[37,38]
*Longidorus* spp. More than 160 species Can transmit Nepoviruses	Polyphagous root-ectoparasites of many plants, including various agricultural crops and trees	Damage is caused by direct feeding on root cells, as well as by transmitting Nepoviruses	Chlorosis and stunted growth in forest trees	[39,40,41,42]
*Pratylenchus* spp. Migratory endoparasites	Possess a wide host range Commonly found in wheat, canola, chickpea, and barley	Provoke plant tissue necrosis because of migration and feeding	Crops show an in-field patchy decline, lack of vigour, chlorosis slower growth, crooked or bushy appearance of tap roots, fleshy tap roots, stunted, stubby small root systems with excessive branching Small roots that are large near the tip Sparse lateral roots Brownish to black spots or streaks or discolored necrotic areas on the roots	[43,44]
*Radopholus* spp. Burrowing nematodes Two species: *R*. *citrophilus* and *R*. *similis*	Affects several economically important crops, e.g., banana citrus, coconut, ginger, palm, avocado, coffee, prayer plant, black pepper, sugarcane, tea, vegetables, ornamentals, trees, grasses, and weeds	Attack the root system Migratory endoparasite in all life stages	In banana, provokes toppling disease In pepper, causes the yellows disease In citrus, can spread decline	[45,46]
*Xiphinema* spp. 39 species have been identified	They have a wide host range that includes common weeds and grasses, strawberries, soybeans, forest trees, orchards, and grapes Can be vectors of viruses, e.g., peach yellow bud mosaic virus in peach, apricot, and plum, cherry rasp leaf virus, and grape yellow vein virus	Attack roots, causing root stunting and tip galling	Necrosis on roots	[47]

**Table 2 pathogens-11-00640-t002:** Advantages and drawbacks of using different strategies of prevention and control of plant-parasitic nematodes.

Prevention/Control Strategy	Advantages	Drawbacks
Chemical control using pesticides	Pesticides occasion a direct lethal effect on the nematodes, and a prompt and effective reduction in the nematode population followed by an improvement in the plant health is expected	(1) Public health risk. The consumption of agricultural food contaminated with pesticide residues shows mutagenic, carcinogenic, cytotoxic, genotoxic, and a range of health-related issues in human beings [56]. Accidental pesticide poisoning can cause a large number of fatalities [57]. (2) Environmental consequences. Contamination of soil and aquifers affects beneficial microbiota, putting soil fertility at risk and enhancing soil erosion [58]. This alteration of the ecosystem could cause an imbalance in flora and fauna population densities with potentially devastating consequences [59]. (3) Using chemical pesticides should be minimised, and their use should be considered only as a part of an integrated control using other sustainable strategies [60].
Crop rotation	The rotation of crops with plants of a different family can reduce the size of nematode populations, thus mitigating their establishment in the new species of plant and reducing the disease [61].	Crops from different families must be alternated, and thus, farmers have to consider changing and alternating their crops.
Planting resistant crop varieties	Using crop varieties with different types of natural genes that cause resistance to nematodes has led to promising results against nematodes [62]. Specialised nematode resistance genes induce active resistance against nematodes and provoke important damage in nematode tissues, including necrosis and the death of nematodes improving the crop health [63].	This system requires RNA technology to select crop varieties with genes associated with resistance to nematodes.
Fallowing	During the off-season, clean fallowing eliminates the nematode plant host availability along with their chance to feed on plants. This simple practice leads to a gradual decline in the nematode population due to nematode deaths because of starvation [18].	None
Soil amendments	Incorporating organic matter, such as compost prepared with animal manure and decomposed plant material, into soil enhances the soil organic matter and proliferation of the microbial biomass, releasing pest-regulating compounds and eventually improving plant health [64].	None
Biological control	The control is highly specific in a blank organism. This practice is the most effective sustainable strategy for the control of plant parasitic nematodes based on the biotechnological use of nematode natural enemies, including fungi, bacteria, and other microorganisms [30,65].	Setting up a biological control system is a costly effort. A lot of planning and money goes into developing a successful system. The time to reduce the parasite population is much slower compared with a chemical pesticide, which produces results immediately.

**Table 3 pathogens-11-00640-t003:** Genera/species of gastrointestinal parasitic nematodes and prevalence recorded in cattle and small ruminants in countries with different climate conditions.

Host	Nematode/Prevalence	Place	Climatic Features	Author
Cattle	*Haemonchus* spp., *Oesophagostomum* spp., *Trichostrongylus* spp., Overall prevalence = 42.33%	Bisofu, Oromia, Ethiopia	warm semi-arid	[83]
	*Haemonchus* spp., *Ostertagia* spp. Overall prevalence = 23.34%	Mosul city, Irak	warm semi-arid	[84]
	Strongylidae order = 16.5% *Strongyloides* 3.8%	Colombian Northeastern Mountain, Colombia	Tropical rainforest	[85]
	*Ostertagia ostertagi* = 41.42%	Germany	Temperate	[86]
	*Strongyloides* spp. = 16.36% *Trichuris* spp. = 22.73%	Kalasin province, Thailand	Tropical savanna	[87]
Sheep	*Chabertia ovina*, *Trichuris ovis*, *Trichostrongylus* spp., *H. contortus* and *Oesophagostomum* spp. Overall prevalence = 36.82%	Assam, India	Tropics	[88]
	Strongylidae order = 31.9%*; Strongyloides* spp. = 3.1% and *Trichuris* spp. = 2.06%	Colombian Northeastern Mountain, Colombia	Tropical rainforest	[85]
Goats	*Trichostrongylus* spp., *Haemonchus* spp. Overall = 88.9%	Coahuila and Nuevo León, (Northeastern Mexico)	Semi-arid	[89]
	*H. contortus* = 97.4%	Maseru, Leshoto, Africa	Mild, warm and temperate	[90]
	*H. contortus* = 47.1%	Bangladesh	Tropics	[91]
	*T. colubriformis, H. contortus, Teladorsagia* spp., *Oesophagostomum* spp., *Trichuris* spp., *Nematodirus spathiger* and *Cooperia curticei* (The whole gastrointestinal tracts of goats at necropsy resulted positive to parasitic nematodes)	Northwest Arkansas, Fayetteville, USA	Warm and temperate	[92]

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
