# Peer review of "Soil-Borne Nematodes: Impact in Agriculture and Livestock and Sustainable Strategies of Prevention and Control with Special Reference to the Use of Nematode Natural Enemies"

_pathogens, 2022, doi:10.3390/pathogens11060640_

Round 1
Reviewer 1 Report
The paper describes in a very reasonable and well organised manner the role of nematodes in plant and animal science, under the agroecological and economic point of view. The author provided a good effort to include a very extensive literature review to approach this subject. There are a few important comments to improve its quality, as follows:
In lines 18-24, the scope of the study is described as the report and description of the problems caused by parasitic nematodes in an agricultural and livestock related point of view, including a special focus to biocontrol. Nevertheless, this is not what the reader expects from the title, which should therefore be modified to reflect the scope of the study.
The abstract has several mistakes and grammatical and syntactic errors in written English. It has to be edited in detail
In line 36, maybe replace “living” with “survival”?
Lines 75-76: please explain and define better ecto- and endoparasitic nematodes
Since the study constitutes a review article, in section 1.2.6, I would suggest to present the control strategies in a Table together with advantages and drawbacks for each one
In lines 215-216, the DNA metabarcoding cannot be considered a cost-effective method for two reasons: (1) high cost and (2) a lot of unnecessary obtained information that cannot be for sure identified at species level due to the relatively small amplicon length
Lines 277-278: please correct the line space
In Figure 4, there are some questionmarks, probably erroneously written. Please correct
Author Response
Reviewer 1
The paper describes in a very reasonable and well organised manner the role of nematodes in plant and animal science, under the agroecological and economic point of view. The author provided a good effort to include a very extensive literature review to approach this subject. There are a few important comments to improve its quality, as follows:
In lines 18-24, the scope of the study is described as the report and description of the problems caused by parasitic nematodes in an agricultural and livestock related point of view, including a special focus to biocontrol. Nevertheless, this is not what the reader expects from the title, which should therefore be modified to reflect the scope of the study.
Author:
Absolutely! I agree with this comment.
I have suggested a new title, trying to reflex the whole content of the paper as follows:
“Soil-borne nematodes: Impact in agriculture and livestock and sustainable strategies of prevention and control with special reference to the use of natural enemies of nematodes”
The new title could be a bit too long; however, I think it covers the whole content of this review in a wider sense, as it is.
Anyway, If Reviewer consider that the title should be shortened, I will be prepared to do it.
Reviewer 1
The abstract has several mistakes and grammatical and syntactic errors in written English. It has to be edited in detail
Author:
The abstract was sent to a company dedicated to improving the written English Language to ensure the quality of our manuscript. If reviewer request, I could supply the English improvement certificate emitted from the English Proof Reading Service company.
Reviewer 1
In line 36, maybe replace “living” with “survival”?
Author:
Ok! The term “living” was replaced. It sounds much better! Thanks!
Reviewer 1
Lines 75-76: please explain and define better ecto- and endoparasitic nematodes
Author:
A brief explanation about ecto and endoparasitic nematodes was provided
Reviewer 1
Since the study constitutes a review article, in section 1.2.6, I would suggest to present the control strategies in a Table together with advantages and drawbacks for each one
Author:
A new table showing advantages and drawbacks in the use of the different control strategies has been created and the text of the section 1.2.6 has been reduced and the new table is referenced.
The new table is shown below:
Table 2 Advantages and drawbacks of using different strategies of prevention and control of plant-parasitic nematodes
Prevention/Control strategy |
Advantage |
Drawbacks |
Chemical control using pesticides |
Pesticides occasion a direct lethal effect on the nematodes and a prompt and effective reduction in the nematode population followed by an improvement in the plant health is expected |
1) Public health risk. The consumption of agricultural food contaminated with pesticide residues shows mutagenic, carcinogenic, cytotoxic, genotoxic, and a range of health-related issues in human beings[40]. Accidental pesticide poisoning can cause a large number of fatalities [41]. 2) Environmental consequences. Contamination of soil and aquifers affects beneficial microbiota, putting soil fertility in risk and enhancing soil erosion [42]. This alteration of the ecosystem could cause an imbalance in flora and fauna population densities with potentially devastating consequences [43]. 3) Using chemical pesticides should be minimized and their use should be considered only as a part of an integrated control using other sustainable strategies [44]. |
Crop rotation |
The rotation of crops with plants of a different family can reduce the size of nematode populations, thus mitigating their establishment in the new species of plant and reducing the disease [45]. |
Crops from different families must be alternated; so, farmers have to consider changing and alternating their crops.
|
Planting resistant crop varieties |
Using crop varieties with different types of natural genes that cause resistance to nematodes has led to promising results against nematodes [46]. Specialized nematode resistance genes induce active resistance against nematodes and provoke important damage in nematode tissues, including necrosis and the death of nematodes improving the crop health [47]. |
This system requires RNA technology to select crop varieties with genes associated with resistance to nematodes. |
Fallowing |
During the off-season, clean fallowing eliminates the nematode plant host availability along with their chance to feed on plants. This simple practice leads to a gradual decline in the nematode population due to nematode deaths because of starvation [18]. |
None |
Soil amendments |
Incorporating organic matter, such as compost prepared with animal manure and de-composed plant material, into soil; enhances the soil organic matter and proliferation of the microbial biomass, releasing pest-regulating compounds and eventually improving plant health [48]. |
None |
Biological control |
The control is highly specific having a blank organism. This practice is the most effective sustainable strategy of control of plant parasitic nematodes based on the biotechnological use of nematode natural enemies, including fungi, bacteria, and other microorganisms [50, 51]. |
Setting up a biological control system is a costly effort. A lot of planning and money goes into developing a successful system. The time to reduce the parasite population is much slower compared with a chemical pesticide that their results are immediately.
|
Reviewer 1
In lines 215-216, the DNA metabarcoding cannot be considered a cost-effective method for two reasons: (1) high cost and (2) a lot of unnecessary obtained information that cannot be for sure identified at species level due to the relatively small amplicon length
Author:
Good point! I agree with you!
The paragraph was modified according to your comment.
Reviewer 1
Lines 277-278: please correct the line space
Author:
The line space was corrected.
Reviewer 1
In Figure 4, there are some questionmarks, probably erroneously written. Please correct
Author:
Dear Reviewer, I am sorry, I don´t see the question marks. May I ask you to help me to identify the written errors, please?

Reviewer 2 Report
Although the topic is very interesting, the review article "Natural nematode enemies as promising biotechnological tools for the prevention of soil-borne parasites" describes in detail the biological properties of various organisms mentioned in the article. I think that the article should be significantly shortened and reshaped because there are too many details that make it difficult to read. One gets the impression that the review is part of the introduction to one of the theses.
part 1.2.1., 1.2.2. and 1.2.3. should be shortened and linked. It is not necessary to cite definitions as a subtitle and then cite them extensively.
it is not necessary to describe in detail all these natural soil borne parasites but to state their potential application. So, the emphasis would be on the potential application of natural enemies of parasites or on the description of their biological properties. In any case, the topic is interesting and after reshaping and removing parts that make it difficult to read, it would be more appropriate.
Author Response
In attention to your valuable comments and suggestions I have shortened the text in Issues 1.2.1, 1.2.2 and 1.2.3 and I have linked these three issues in only one, such as you kindly suggested. The modified text is marked in green color. It is worth mentioning that Reviewer 1 suggested to modify the title of the paper to reflex the content of this review. The new proposed title is: “Soil-borne nematodes: Impact on Agriculture and Livestock and sustainable strategies of prevention and control with special reference to the use of nematode natural enemies”. So, the descriptions of some natural soil-borne parasites stayed as they were. Unless you suggest to delete this information. Thank you very much for your valuable comments.
Round 2
Reviewer 2 Report
The author accepted the suggestions and I have no further comments.